# Metallo-supramolecular branched polymer protects particles from air-water interface in single-particle cryo-electron microscopy

Yixin Xu [1,6,7], Yuqi Qin [1,7], Lang Wang [2], Yingyi Zhang [3], Yufeng Wang [2✉] & Shangyu Dang [1,4,5✉]

Recent technological breakthroughs in single-particle cryo-electron microscopy (cryo-EM) enable rapid atomic structure determination of biological macromolecules. A major bottle-neck in the current single particle cryo-EM pipeline is the preparation of good quality frozen cryo-EM grids, which is mostly a trial-and-error process. Among many issues, preferred particle orientation and sample damage by air–water interface (AWI) are common practical problems. Here we report a method of applying metallo-supramolecular branched polymer (MSBP) in the cryo-sample preparation for high-resolution single-particle cryo-EM. Our data shows that MSBP keeps a majority of particles away from air–water interface and mitigates preferred orientation as verified by the analyses of apoferritin, hemagglutinin) trimer and various sample proteins. The use of MSBP is a simple method to improve particle distribution for high-resolution structure determination in single-particle cryo-EM.

[1] Division of Life Science, The Hong Kong University of Science and Technology, Clear Water Bay, Hong Kong, China. [2] Department of Chemistry, The University of Hong Kong, Hong Kong, China. [3] Biological Cryo-EM Center, The Hong Kong University of Science and Technology, Clear Water Bay, Hong Kong, China. [4] Southern Marine Science and Engineering Guangdong Laboratory (Guangzhou), Guangzhou, China. [5] HKUST-Shenzhen Research Institute, Nanshan, Shenzhen, China. [6] Present address: Department of Biology, Institute of Molecular Biology and Biophysics, ETH Zurich, Zurich, Switzerland. [7] These authors contributed equally: Yixin Xu, Yuqi Qin. ✉email: wanglab@hku.hk; sdang@ust.hk

Visualization of biological macromolecules at atomic resolution provides valuable information for unravelling the fundamental mechanisms of many biological processes and also for driving the development of drugs for diseases caused by dysfunctional biological macromolecules. Recent technical breakthroughs in single-particle cryo-EM have powered a "resolution revolution" in structural biology. Cryo-EM circumvents the major challenges faced when using traditional X-ray crystallography method[1–4]. Despite the advances, there are still some recurring problems limiting the application of cryo-EM. Among these challenges, preparations of cryo-specimen that are suitable for high-resolution 3D reconstruction using single-particle cryo-EM is still a bottleneck in many cases[5]. For some proteins, this is the most difficult obstacle to overcome, as a large proportion of protein particles tend to adsorb to the air-water interface (AWI) in grids, which causes particle disruptions and introduces preferred orientations, making high-resolution structure determinations challenging.

Air-water interface is the main factor causing serious problems in sample preparation because the rate of particles hitting the AWI is much faster than the vitrification of specimens by plunge freezing[6]. By investigating particle distribution in vitreous ice for more than 1000 single particle tomograms, a previous study shown that the vast majority of particles (about 90%) are adsorbed to AWI[7], signifying a common problem in cryo-specimen preparation. As a result, the particles adsorbed and trapped in the AWI may partially unfold and cause structural deformation or even denaturation[8]. Another common situation is that the particles on the AWI exhibit strong orientation preference[9]. Insufficient native conformation protein particles in random orientations make reconstructing a reliable three-dimensional structure at high resolution difficult if not impossible[6].

Several methods have been developed to improve particle behavior in vitreous ice. In some cases, adding detergent, like CHAPSO, can eliminate particle adsorption to AWI[10]. Coating of cryo-EM grids with a thin layer of graphene materials as supporting film also improves sample quality in vitreous ice[11–13]. Furthermore, the effect of AWI can be reduced by using a rapid plunge-freezing robot, Chameleon, by decreasing the plunge freezing time compared with conventional vitrification devices[14–16]. In addition, collecting tilted images can also address the insignificant preferred orientation problem[17]. However, the need for special devices or complicated techniques limits the wide application of these methods.

Here we report on the successful application of a palladium-polyethylene glycol (PEG) MSBP system in the sample preparation process, which significantly reduces the AWI effect. The MSBP is formed by functionalizing the chain ends of the PEG with bispyridyl ligands and introducing palladium ions (Pd$^{2+}$) at room temperature (Fig. 1a). The system was previously shown to form hyper-branched polymers[18], which may potentially surround the proteins in solution and shield the particles away from the two AWIs on both sides of the grid.

The system involves simply mixing the polymer ligands and metal ions as an additive, which is introduced to protein particles of interest (Fig. 1b). We first confirmed that proteins of different sizes prepared using MSBP can be resolved at a high resolution by single-particle cryo-EM method. We then showed by using cryo-ET technique that protein particles with MSBP are uniformly distributed in the inner layer of the vitreous ice without assembling at the AWI as observed for proteins without MSBP. In addition, we showed that application of MSBP improved the angular distribution of particles used for 3D reconstruction in several cases. This attribute is best illustrated in the case of hemagglutinin (HA) trimer resolved to 3.19 Å by minimizing the preferred orientation problem known for HA trimer[14]. Our study suggests that applying MSBP is a straightforward and general

method for improving particle distribution and orientation of cryo-specimen in vitreous ice for single-particle cryo-EM study.

## Results

**MSBP does not affect high-resolution structure determination.** To ensure the practicality and biocompatibility of our approach for high-resolution structure determination using single-particle cryo-EM, we prepared cryo-specimen of apoferritin following the standard protocol except adding MSBP. Briefly, apoferritin (MW: 504 kDa) was mixed with bispyridyl PEG (MW: 5000 in this study) and palladium nitrate; the mixture was then immediately applied to cryo-grids and incubated for several minutes to allow formation of metallo-supramolecular structure before blotting and plunge freezing (Fig. 1b). The concentration of MSBP has been optimized to ensure proper ice thickness and reduce background noise caused by the metallo-supramolecular structure. Although the concentration of the polymer (2 mg/ml) used to form a branched architecture is well below the gel point (about 30 mg/ml), changes in the solution physical properties, especially the viscosity, are obvious (Supplementary Video 1). The raw image of apoferritin prepared with MSBP (Fig. 1c) shows homogenous and monodisperse apoferritin particles without obvious background noise that may be contributed by the polymer. In addition, the overall ice quality in grids prepared using MSBP is better than that in grids without application of MSBP as indicated by estimated resolution in contrast transfer function (CTF) determination (Supplementary Fig. 1a). This result indicates that the metallo-supramolecular structures may influence the AWI to improve reproducibility of cryo-grids by generating better vitreous ice. The good particles of apoferritin, sorted out by 2D and 3D classification, were used to generate 3D reconstructions. The final resolution, calculated using gold-standard Fourier shell correlation (FSC), reaches 2.16 Å, very close to the Nyquist limit of acquired micrographs (Fig. 1d, Supplementary Fig. 1, Supplementary Table 1). Local resolutions show high qualities that enable the fitting of individual residues from the published structure (PDB: 7KOD) to the density map unambiguously (Fig. 1e). We have also applied MSBP to other proteins of different molecular weights. Application of MSBP did not affect the high resolution structure determination of catalase (MW: 240 kDa) (Supplementary Fig. 2a, b, Supplementary Table 1) and β-galactosidase (MW: 464 kDa) (Supplementary Fig. 2c, d, Supplementary Table 1), suggesting the biocompatibility and practicality of MSBP to facilitate single–particle cryo-EM studies. These results showed that the application of MSBP in sample preparation of single-particle cryo-EM can achieve near-atomic resolution of proteins with different sizes.

To evaluate the influence of MSBP, we collected protein alone datasets under similar conditions and performed a systematic comparison with datasets of protein with MSBP applied for these three proteins. Interestingly, for catalase and apoferritin, the overall resolution of density maps with MSBP is slightly (but not significantly) better compared to those without MSBP (Supplementary Fig. 2a, Supplementary Fig. 3), probably because of the improved ice quality by introduction of MSBP. We further studied the relationship between particle numbers and resolution for apoferritin. For each dataset, particles used for final 3D reconstruction were split into several 50k particle groups randomly and performed 3D refinement with one or combined groups. Consistently, apoferritin with MSBP resolves a density map with slightly higher resolution using the same number of particles (50k, 100k, and 150k), further proving the introduction of MSBP has almost no negative effect on final resolution (Supplementary Fig. 3a). Surprisingly, the refinement of CTF (global and local) significantly improved resolution of apoferritin

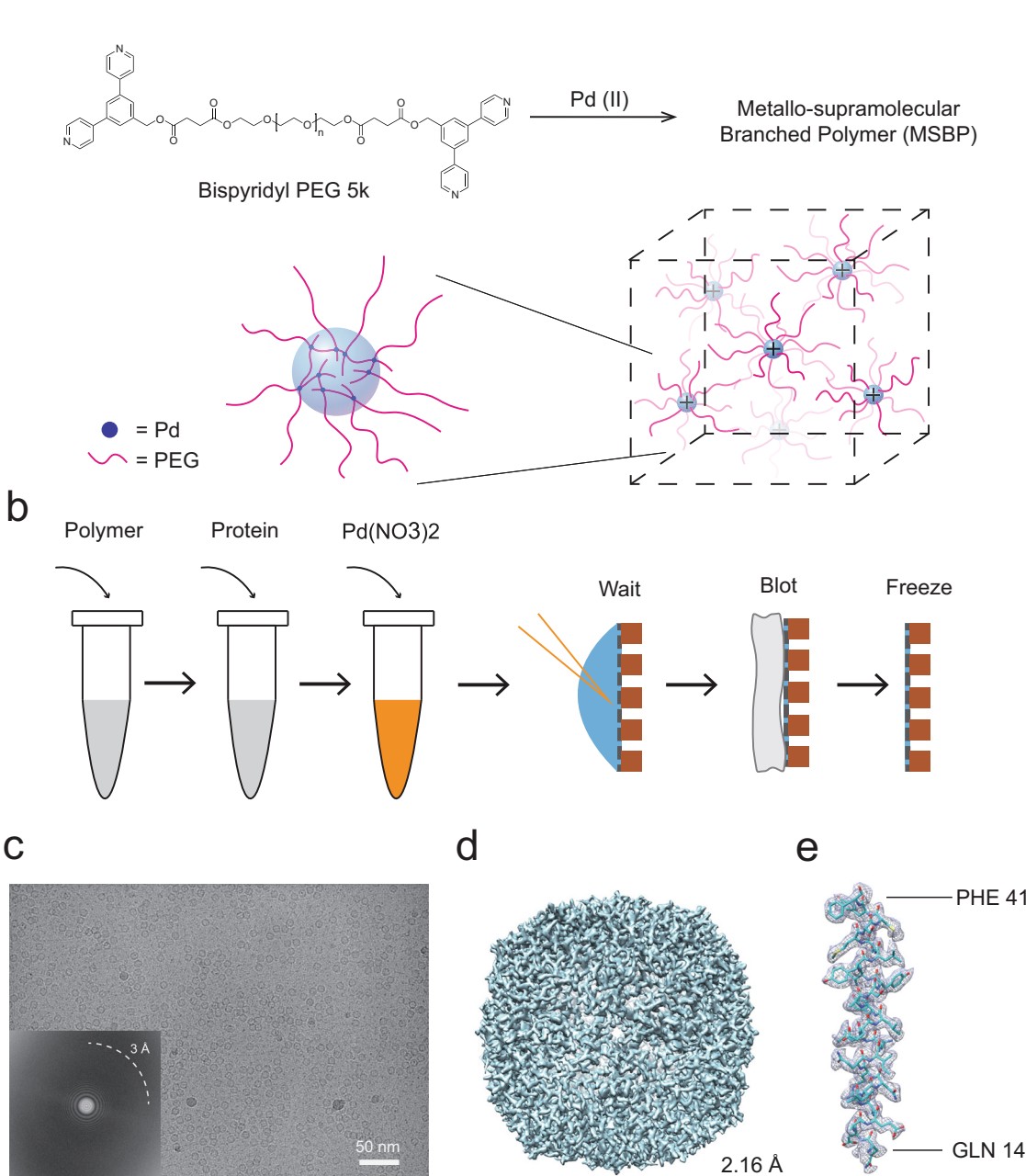

**Fig. 1 Apoferritin, with application of metallo-supramolecular branched polymer (MSBP) in sample preparation, can achieve a high resolution.**
**a** Schematic diagram of local structure of MSBP in solution. Branched polymers are formed in solution. **b** Workflow for preparation of apoferritin cryo-specimen with MSBP. Apoferritin, mixed with modified PEG5000 and $Pd(NO_3)_2$, was loaded onto the holey carbon grid. **c** A representative cryo-micrograph of apoferritin prepared with MSBP. Fourier transformation of the micrograph is also shown in left bottom with Thon rings extending to 3 Å. **d** 3D reconstruction of apoferritin prepared with MSBP at 2.16-Å resolution. **e** Representative cryo-EM densities of one α helix of apoferritin. The density is shown as grey mesh and the corresponding structural model (PDB: 7KOD) in blue with side chains of residues depicted to demonstrate the quality of the map.

with MSBP compared to apoferritin alone, mainly through local refinement (Supplementary Fig. 3b). MSBP may achieve a more evenly distributed ice thickness and push most particles into the central layer of the vitreous ice, while apoferritin-only particles are primarily distributed near the AWI (Supplementary Fig. 4a, b). The large range of defocus in apoferritin particles with MSBP (Supplementary Fig. 3c) will be further optimized through CTF refinement, thereby contributing to significant resolution improvement. Together, our systematical assessment

demonstrated application of MSBP will not affect, but may benefit in some cases, high-resolution structure determination in single-particle cryo-EM.

**Background noise introduced by MSBP can be neglected.** High-resolution structure determination of catalase (MW: 240 kDa for homotetramer) has demonstrated that background noise introduced by MSBP has little effect on protein with medium size (Supplementary Fig. 2a, b). However, the influence could be

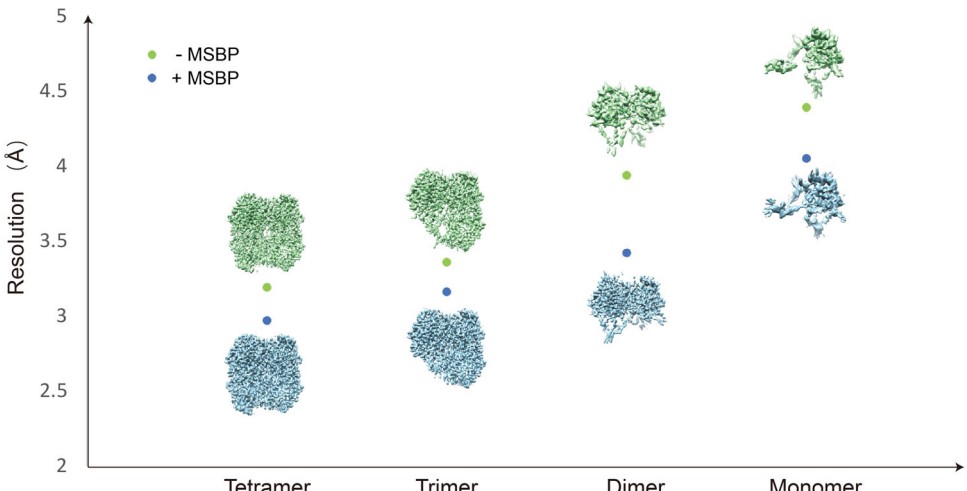

**Fig. 2 Background noise introduced by MSBP could be neglected.** Comparison of resolution for tetramer, trimer, dimer, and monomer of catalase reconstructed using the same number of subtracted particles. The maps of catalase with or without MSBP are shown as blue and green, respectively. The blue and green dots indicate resolution of corresponding maps with or without MSBP respectively.

significant for small proteins. Based on the dataset of catalase with MSBP, we performed a systematic analysis to evaluate applicability of MSBP to small proteins by using particle subtraction. Unsurprisingly, the resolution of the catalase map reconstructed using catalase particles with the same number is gradually dropped from trimer (MW: 180 kDa), dimer (MW: 120 kDa) to monomer (MW: 60 kDa) of catalase (Fig. 2).

To check if the resolution drop caused by MSBP could be compensated by increasing particle number for small proteins, we performed symmetry expansion before particle subtraction to generate more particles for monomer catalase. As expected, the more particles included for reconstruction, the higher resolution can be achieved for catalase monomer (Supplementary Fig. 6). The resolution of catalase monomer was improved from 4.05 Å to 3.36 Å by using four times the particle number, further proven by improved map quality for confidential model building. We also analyzed catalase only dataset following the same strategy, which shows comparable resolution decrease to that of catalase with MSBP (Fig. 2). These results implied that the background noise introduced by MSBP has little effect on cryo-EM structural studies for small proteins, and the potential resolution loss could be compensated by collecting a larger dataset if needed.

**MSBP improves particle distribution in vitreous ice.** To further investigate if MSBP will improve particle distribution in vitreous ice by preventing attraction of particles into the AWI during plunge freezing, we used cryo-ET to locate individual particles in vitreous ice[14]. We first collected cryo-ET dataset for apoferritin with or without MSBP for comparison (Supplementary Videos 2, 3) and picked individual particles automatically by template-matching from reconstructed tomograms. For apoferritin without MSBP, an abundance of apoferritin particles were trapped in the AWI, and only a small number of particles were found staying in the middle section of the tomogram (Fig. 3a, b, Supplementary Fig. 4a). In contrast, for apoferritin treated with MSBP, particles distributed uniformly throughout the vitreous ice layer with few particles appearing at the AWI (Fig. 3c, d, Supplementary Fig. 4b). These distinct particle distributions are more obvious when compared in side views (Y-Z) of the tomograms (Supplementary Fig. 5a, b). Also, the obvious air-water interfaces, indicated by the ice contaminations, showed a thinner ice layer for samples treated with MSBP. We also collected and analyzed cryo-ET datasets of apoferritin treated with PEG or palladium nitrate

only, both showing similar particle distribution as the dataset of apoferritin only (Supplementary Fig. 4d, e, Supplementary Fig. 5d, e).

To check if this is a general effect of MSBP, we also applied cryo-ET studies for other proteins, including HA, catalase, β-galactosidase, two intramembrane proteins (IMP1 and IMP2), and CSW (C9ORF72-SMCR8-WDR41) complex. Consistently, most of the protein particles are protected from the AWI by applying MSBP, while particles are trapped on AWI without MSBP (Fig. 3e, Supplementary Fig. 7-12, Supplementary Video 5), although the efficiency varies among different proteins. These results showed that application of MSBP might be a general solution to improve particle distribution in an even thinner vitreous ice by keeping particles away from AWI.

**MSBP alleviates preferential orientations.** We investigated whether particle orientations in vitreous ice are also changed by keeping away from AWI. To test this hypothesis, we collected and analyzed single-particle cryo-EM datasets for hemagglutinin trimer treated with or without MSBP. Consistent with previous studies[14,19], the cryo-EM density map reconstructed using HA trimer only dataset showed distorted feature, indicating severe preferred orientation problem of the data. In contrast, the application of MSBP to HA trimer generated a high-resolution map at 3.19 Å that can fit the structure (PDB: 6WXB) very well (Fig. 4a, Supplementary Fig. 13a, b, Supplementary Table 1). To evaluate the map quality, we performed directional FSC (dFSC) analysis for both maps of HA trimer reconstructed from dataset with and without MSBP (Fig. 4b). For HA trimer only, dFSC showed a wider resolution distribution, suggesting some views were missing in particles used for final reconstruction. For HA trimer with MSBP, a more concentrated resolution distribution of dFSC curves indicated particles used for final reconstruction were almost equally distributed in different angular directions (Fig. 4b, Supplementary Fig. 13c). This result further confirmed that preferred orientation problem of HA particles in vitreous ice is minimized by introducing MBSP in cryo-sample preparation. Moreover, catalase and β-galactosidase reconstructed using dataset with MSBP also show more even angular distributions compared to maps from dataset without MSBP (Fig. 4c), suggesting particles adopt a more random orientation in vitreous ice when protected from AWI by MSBP.

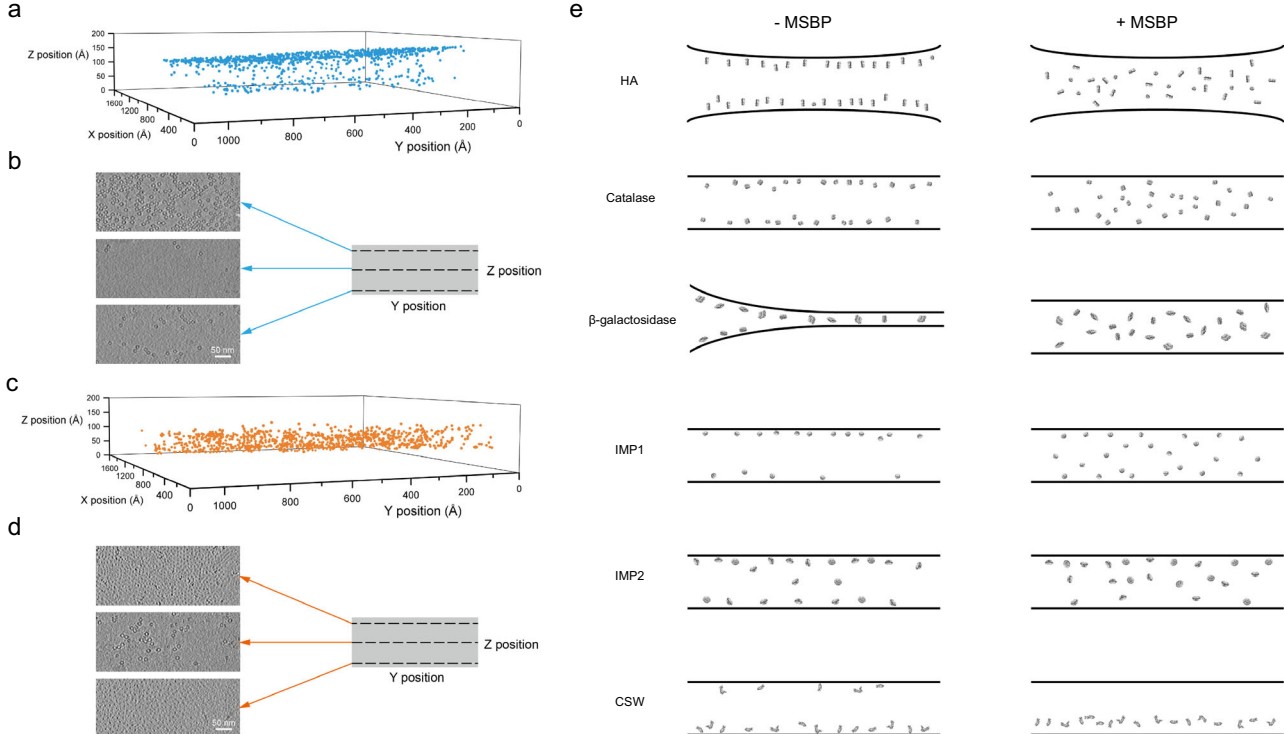

**Fig. 3 MSBP improves protein particle distribution in vitreous ice analyzed by cryo-ET. a** Localization of particles from apoferritin without MSBP grid shows most of particles were adsorbed to the top air-water interface. Particles were picked by template matching in Dynamo. **b** Particle distribution of apoferritin without MSBP in three layers along the Z axis of the tomogram. **c** Localization of particles from apoferritin with MSBP grid shows more uniform distribution. Particles were picked by template matching in Dynamo. **d** Particle distribution of apoferritin with MSBP in three layers along the Z axis of the tomogram. Regular arranged small particles are observed in two air-water interfaces. **e** Particle distribution of investigated proteins in vitreous ice without (left) or with (right) MSBP.

To further assess the angular distribution of HA particles in vitreous ice, we performed 2D classifications and quantitative analyses of particles with different projection views. 2D averages of HA trimer particles showed three distinct orientations: top view, side view, and tilted view (Supplementary Fig. 13d, e). We then calculated the number of particles grouped into each of these three categories for the two respective datasets (Fig. 4d). The dataset of HA trimer without MSBP exhibited an obvious preferred orientation problem, with more than 50% of particles oriented in top views while only about 13% particles showing typical side views, consistent with previous studies[14,20]. For HA trimer treated with MSBP, the proportion of particles representing top views decreased dramatically (from 57.4% to 23.9%), while particles from both side and tilted views increased correspondingly. It is also worth noting that particles are almost proportionately distributed into three different views in dataset of HA trimer treated with MSBP. 2D averages of top or side views both contain about 30% of total particles, while more than 40% of particles are presented as tilted views, the largest category in this dataset. Comparing the statistics of the two HA datasets under different conditions clearly shows that the application of MSBP can alleviate the preferred orientation problem of HA trimmer by protecting particles from AWI such that particles can rotate and distribute more freely in vitreous ice.

## Discussion

In summary, we developed a simple and promising method to improve particle behavior in vitreous ice by applying the MSBP for high-resolution single-particle cryo-EM structure determination. We have shown that involvement of MSBP in cryo-specimen preparation reduces particles trapped at AWI, thus alleviates the preferred orientation problem by producing more randomly distributed particles in vitreous ice. Although further investigation needs to be performed, protecting particles away from AWI by MSBP may prevent denaturing, partially denaturing, and disassembly of proteins and protein complexes, which are usually caused by adsorption of particles onto AWI[21]. In addition to other reported methods such as modification of cryo-grids with graphene oxide, the simplicity of our approach provides an alternative strategy to optimize cryo-sample preparation conditions for challenging protein complexes that do not behave optimally on holey carbon grids.

As far as we know, this is the first report of applying metallo-supramolecular polymers for sample preparations in cryo-EM study. Our MSBP consists of chain-end-modified PEG and palladium ions. PEG is a well-established biological friendly reagent that rarely introduces artificial structures. It has many applications in the biology field, such as facilitating cytoskeletal organization of osteoblasts in bone tissue engineering[22] and inhibition of cell migration[23]. There are also ample studies using photo-affinity palladium reagents to study protein-protein interactions[24], indicative of the biocompatibility of palladium. The formation of the MSBP can withstand extreme pH and salt conditions, thus have a wide application potential for biological macromolecules in general. Indeed, MSBP can protect particles of catalase away from AWIs even in a high salt concentration (1 M NaCl), showing similar effect as it does under normal salt condition (50 mM NaCl) (Supplementary Fig. 14). Although the probability is very small, more studies need to be performed to see if MSBP would cause protein damage or change protein structures in future.

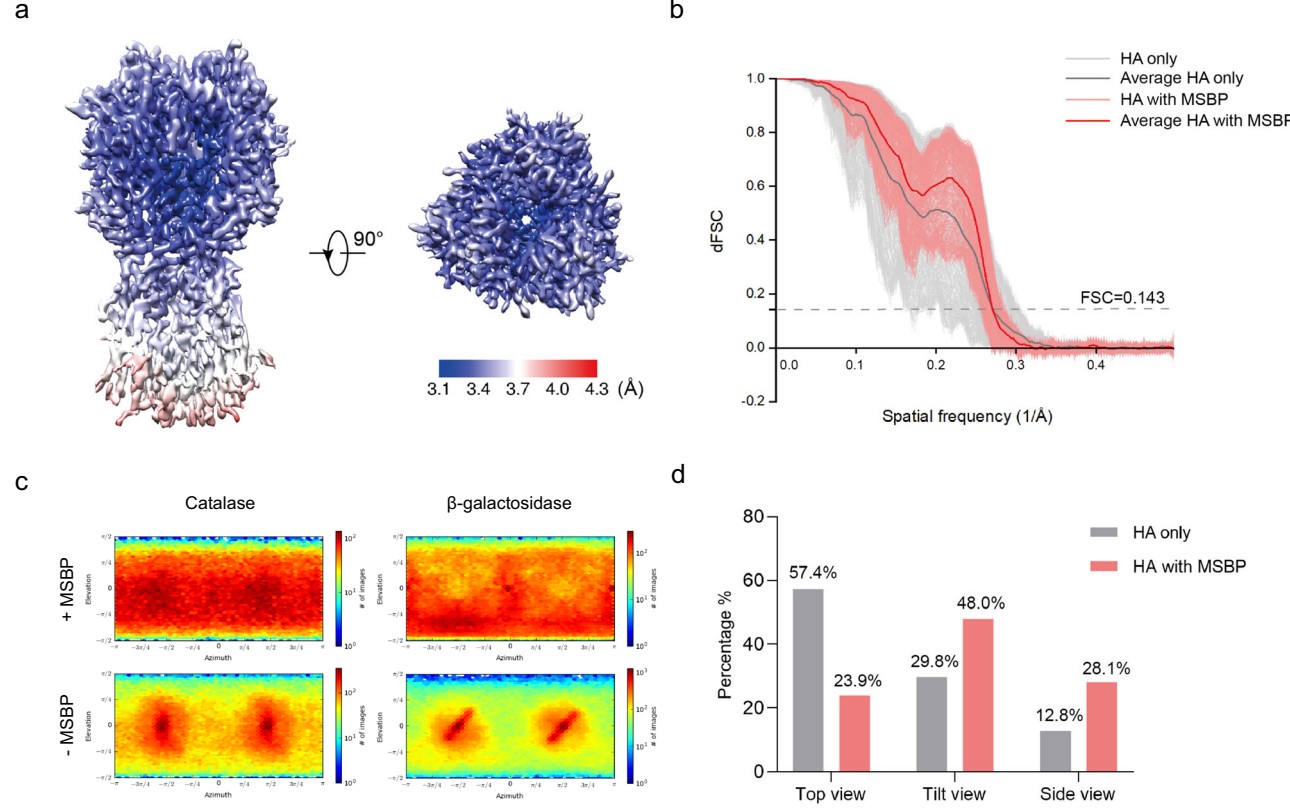

**Fig. 4 Cryo-EM analysis of hemagglutinin (HA) trimer. a** Local resolution of HA trimer in two different views, as estimated by cryoSPARC and shown with pseudo-color representation of resolution. **b** dFSC from different Fourier cones for HA with or without MSBP. Each curve indicates a different direction. **c** Angular distribution of reconstructed catalase (left) and β-galactosidase (right) with (top) or without (bottom) MSBP. **d** Statistics of HA trimer particles in three different views. Obviously, top views are the preferred orientation of HA trimer without MSBP. The proportions of the side views and tilted views of HA trimer with MSBP increased significantly.

The existing solutions to address AWI caused problems could be classified to three directions roughly. The first strategy is optimizing interaction of particles and AWI by changing properties of particle itself or AWI, exemplified by condition screening for sample preparation and adding additives like detergents. The second one is to avoid AWI by introducing solid-water interface, such as coating holy carbon grids with graphene or graphene oxide, and recently developed nanofluidic chip[25]. The third strategy is to fast freeze the protein particles before they adsorb to AWI by using fast freezing devices. Although future studies are needed to understand the exact nature of how MSBP works, MSBP may benefit cryo-sample preparation through multiple mechanisms as outlined below.

First, MSBP may occupy the AWI therefore shielding protein particles away from the AWI. This hypothesis is supported by the observation of the network formed by plenty of uniformly arranged small polymeric particles in the AWIs of cryo-grid prepared using MSBP (Supplementary Videos 3, 4, Supplementary Fig. 4b, c, Supplementary Fig. 5b, c). We did not observe similar networks in cryo-grids without addition of MSBP (Supplementary Video 2, Supplementary Fig. 4a, 5a), suggesting the network is formed by the MSBP. Although MSBP shares similar effect by occupying AWIs with detergent[8], the mechanism is different. The occupancy of MSBP in AWI may achieved by attraction of positively charged MSBP to the negatively charged AWI[26], which are different from detergents occupying AWIs by forming an amphiphilic monolayer[27].

Second, we hypothesize that the presence of the branched polymer in the solution may change the local viscosity of the solution[28] and retards the adsorption rate of the protein into the AWI (Supplementary Video 1). Although different from high-speed freezing[14], our method shares the same idea to reduce spread rate of particles to AWI by increasing solution viscosity. Alternatively, the MSBP may be recruited to the AWI with a speed faster than protein particles and form a network that protects particles from adsorption into AWI.

Besides the two potential mechanisms aforementioned, most likely, MSBP could affect the adsorption propensity of the protein to the AWI[29] by changing surface properties of the protein, particular distribution of charge and hydrophobicity. Previous study has shown that PEGylation, covalently link modified PEG molecules to exposed primary amines, could retain protein particles in the central layer of the ice by forming a shielding mask on protein surface[30]. Here, MSBP is designed as a composite nanocluster that combines hydrophilic polymers and charged metal ions in a core-shell structure. Binding of MSBP will generate a "homogenized" surface of proteins by making them isotropic and hydrophilic, so that the protein particles can remain in the aqueous phase and later in the vitreous ice. The positively charged MSBP nanoclusters can weakly attach to negatively charged region in protein surface via Coulombic interaction. The hydrophilic PEG polymers surround the proteins thus protecting them from hitting the AWI and staying in solution rather than AWI (Supplementary Fig. 15). In addition, due to steric repulsion provided by the PEG polymers, MSBP stays loosely bound to the protein particles, therefore has little chance of influencing protein structures.

Although application of MSBP for cryo-sample preparation would introduce some background noise in collected micrographs and provide a smaller number of particles due to a greater spatial distribution available for protein molecules (Supplementary

Fig. 16, 17), the final 3D reconstruction is affected little since these random background noises could be averaged out by data processing. For protein with large and medium mass, like apoferritin and catalase in this study, the 3D reconstruction actually is benefited by application of MSBP, giving a higher reported resolution for cryo-EM density map generated using the same number of particles (Supplementary Fig. 2a, b, Supplementary Fig. 3a, b), which further suggest association of protein particles onto AWI may cause instability or even damage of the protein. The systematic studies of catalase using subtracted particles also suggested that for small proteins, the influence of background noise caused by MSBP could be negligible, although further studies are required for other proteins with small mass using intact particles. For proteins with small mass like HA trimer, the resolution is affected by these background noises, due to a lower signal-to-noise ratio (SNR) in micrographs with application of MSBP. However, this problem could be compensated by collecting a larger set of particles, to strengthen SNR and further improve resolution. Using a larger dataset for HA with MSBP, we finally obtained a high quality cryo-EM density map for structural analysis (Fig. 4a, Supplementary Fig. 13a, b).

In some cases, addition of detergent can also improve particle distribution in vitreous ice[10]. Systematical study shows nonionic and zwitterionic but not ionic detergent could separate protein particles from AWI[31]. Specifically, addition of low concentration cetyltrimethylammonium bromide (CTAB), a cationic surfactant, to cryo-sample of HA trimer could provide more evenly distributed protein particles to facilitate high-resolution structure determination, although most of the protein particles were still adsorbed to AWI[31]. Different from surfactant that has high affinity for AWI and exposes their hydrophobic tail outside, MSBP is highly hydrophilic and has a more random distribution in solution. In addition, MSBP increases viscosity of the solution, which might be one of the reasons for providing a random distribution of protein particles inside. Furthermore, addition of detergent to improve protein angular distribution is also a trial-and-error process. For proteins other than HA trimer, introducing CTAB limited particle distribution in a narrower angular range than without CTAB[31]. Comparison of particle distribution for HA trimer with or without MSBP further demonstrated our method can facilitate an even distribution of protein particles in vitreous ice with a different mechanism than that using surfactant (Supplementary Fig. 7).

Protecting biological macromolecules from AWI is a reasonable way to avoid recurring problems in cryo-specimen preparation for single particle cryo-EM studies. Previously, different methods have been developed to address AWI problem, including tracking proteins away from AWI using graphene oxide based substrate approach[12], changing protein surface properties by PEGylation[30,32], and so on. In the long term, we aim to develop a universal method that can be applied to most of the biological macromolecules, despite the differences they may have. In this study, we introduced MSBP in cryo-sample preparation to reduce the probability of particles interacting with AWI. Although the concentration we used is considerably lower than the required concentration for gel formation, the physical properties of the solution are changed apparently (Supplementary Video 1). Following this same principle, if we can identify reagents that could form biological compatible branched structures or network (either chemical or protein network) at an even lower concentration, their ability to protect protein particles away from AWI would increase dramatically compared to MSBP. Such a network is more ideal for application to sample preparation in single particle cryo-EM to minimize AWI effects. Inside of the network, particles are surrounded randomly, thus the potential noise introduced by network certainly could be averaged out

during data processing and would affect little of the final map quality. We will continue to search for polymer networks with the desired properties that can improve cryo-sample preparations for single particle cryo-EM studies.

Briefly, our approach described here will help cryo-sample preparation by diminishing AWI effects with a wide applicable potential, eventually benefiting the whole cryo-EM community.

## Methods

**Synthesis of bispyridyl polyethylene glycol (bispyridyl PEG).** To a 50 mL Schlenk flask equipped with a magnetic stirring bar were added [3,5-di(pyridine-4-yl)phenyl]methanol (2.5 g, 9.54 mmol), succinic anhydride (1.15 g, 11.5 mmol), and 4-dimethylaminopyridine (1.4 g, 11.5 mmol). The flask was capped with a septum and evacuated and refilled with nitrogen three times. To the flask via syringe were added 30 mL of anhydrous dichloromethane (DCM) under nitrogen. The flask was placed at room temperature for overnight. The contents of the flask were concentrated via rotary evaporation to give crude product which was subjected to chromatography on silica gel ($CH_2CL_2$/MeOH/ $CH_3COOH$ step gradient, $10:0:0 \rightarrow 97:1.5:1.5 \rightarrow 94:3:3$). The fractions containing desired product were combined, concentrated to yield a light yellow solid. The product was further purified by washing with de-ionized water for 8-10 times to remove the excess of succinic acid and acetic acid, and dried by lyophilization to give off-white powdery solid. HRMS (EI): calcd. for $C_{21}H_{18}O_4N_2$, most abundant m/z = 362.1261; found, 362.1270.

Polyethylene glycol was then coupled with the succinylated ligand via a DIC ($N,N'$-diisopropylcarbodiimide)-mediated coupling protocol. Typically, to a 25 mL Schlenk flask equipped with a magnetic stirring bar were added the OH-$PEG_{5k}$-OH (300 mg, 0.06 mmol), the succinylated bispyridyl ligand (87 mg, 0.24 mmol), and 4-dimethylaminopyridine (8.0 mg, 0.065 mmol), the flask was capped with a septum and evacuated and refilled with nitrogen three times. To the flask via syringe were added 10 mL anhydrous dichloromethane (DCM) and 2 mL anhydrous $N,N$-dimethylformamide (DMF) under nitrogen. DIC (80 µL, 0.52 mmol) was then added under nitrogen. The flask was placed at room temperature for 48 h. The crude product was purified by prep-HPLC, eluent flow rate was 10 mL/min, and the eluent composition consisted of mixtures of ultrapure water and HPLC grade acetonitrile. The eluent gradient consisted of a linear ramp from 35% to 80% acetonitrile during 0-30 min, followed by a ramp to 95% acetonitrile during 30-45 min. Fractions containing the product were combined, frozen and lyophilized.

**Cryo-grids sample preparation.** The mouse heavy chain apoferritin sample was expressed and purified following the standard protocol[33].

Bispyridyl PEG-5k (L-PEG-L) was dissolved to 50 mg/ml in $H_2O$, and Pd($NO_3)_2$ was dissolved to 5 mg/ml in DMSO as stock solution. The stock solution was further diluted to the indicated concentrations using the same buffer as proteins before mixing with protein sample. The holey carbon grids were glow-discharged for 30 s before application of all cryo-specimen used in this study.

For single-particle cryo-EM grid, 1 µL apoferritin (2 mg/ml), 1 µL L-PEG-L (6 mg/ml) and 1 µL Pd($NO_3)_2$ (0.72 mg/ml) were well mixed and applied to holey carbon grids. After 60 s incubation on the grids at 22 °C under 100% humanity, the grids were blotted with filter paper (TED PELLA 595) for 6 s and plunge-frozen into liquid ethane cooled by liquid nitrogen using a FEI MarkIV Vitrobot. To prepare cryo-grids for cryo-ET studies, a similar protocol was used except gold tracers (BSA tracer Conventional, 10 nm, AURION) were mixed with sample before loading to the

grids. The MSBP grids were prepared by replacing apoferritin with buffer. To prepare the cryo-grid of apoferritin without MSBP, 3 µL apoferritin (0.67 mg/ml) was loaded on the holey carbon grids (Quantifoil 400 mesh Cu R1.2/1.3) which was glow discharged for 30 s. The grid was plunge-frozen by the same protocol.

The HA trimer recombinant protein purchased from MyBio-Source (Catalog number MBS434205) was dissolved in PBS buffer to indicated concentration. To prepare HA trimer only grid for single-particle cryo-EM studies, 3 µL HA trimer (0.75 mg/ml) was applied to holey carbon grid (Quantifoil 400 mesh Au R1.2/1.3) which was glow discharged 30 s at 15 mA. After 30 s incubation at 22 °C under 100% humanity, the grid was blotted with filter paper (TED PELLA 595) for 4 s and plunge-frozen into liquid ethane cooled by liquid nitrogen. To prepare HA trimer with MSBP grid for single-particle cryo-EM studies, 3 µL HA trimer (1 mg/ml), 0.5 µL L-PEG-L (8 mg/ml) and 0.5 µL $Pd(NO_3)_2$ (1 mg/ml) was applied to holey carbon grid (Quantifoil 200 mesh Cu R2/2) which was glow discharged 45 s at 15 mA. After 120 s incubation at 22 °C under 100% humanity, the grid was blotted with filter paper (TED PELLA 595) for 7 s and plunge-frozen into liquid ethane cooled by liquid nitrogen. To prepare HA trimer only grid for cryo-ET studies, 3 µL HA trimer (0.75 mg/ml) was applied to holey carbon grid (Quantifoil 400 mesh Cu R1.2/1.3) which was glow discharged 30 s at 15 mA. After 60 s incubation at 22 °C under 100% humanity, the grid was blotted with filter paper (TED PELLA 595) for 4 s and plunge-frozen into liquid ethane cooled by liquid nitrogen. To prepare HA trimer with MSBP grid for cryo-ET studies, a similar protocol was used except 0.5 µL L-PEG-L (4 mg/ml) and 0.5 µL $Pd(NO_3)_2$ (0.5 mg/ml) were mixed with 3 µL HA trimer (1 mg/ml) for grid loading.

The catalase purchased from Sigma (catalog number: C3556) was applied to holey carbon grid (Quantifoil 400 mesh Cu R1.2/1.3) with a final concentration of 1 mg/ml. The β-galactosidase purchased from Sigma (catalog number: 10105031001) was dissolved to 3 mg/ml with buffer containing 25 mM Tris (pH8.0), 50 mM NaCl, 2 mM $MgCl_2$, 2 mM β-mercaptoethanol, and applied to holey carbon grid (Quantifoil 400 mesh Cu R1.2/1.3). 3 µL catalase or β-galactosidase were mixed with 0.5 µL L-PEG-L (16 mg/ml) and 0.5 µL $Pd(NO_3)_2$ (final concentration of MSBP is 2 mg/ml) was used to prepare cryo-sample following the same protocol as that for hemagglutinin.

**Data collection and processing**. All cryo-EM data collection was done with an FEI Titan Krios G3i electron microscope (Thermo Fisher Scientific) equipped with a high-brightness field emission gun operated at 300 kV.

For single-particle cryo-EM, images were recorded with a K3 Summit direct electron detector (Gatan) using EPU (Thermo Fisher Scientific) in counting mode at a calibrated magnification of 81,000X (1.06 Å physical pixel size). The slit width of the Gatan Imaging Filter (GIF) Bio Quantum was set to 20 eV. 2090 micrographs were collected over 8 s with 7.13 e⁻/Å²/s dose rate for the dataset of apoferritin. 1587 micrographs were collected over 8 s with 7.13 e⁻/Å²/s dose rate for the dataset of apoferritin with MSBP. 576 micrographs were collected over 5 s with 10.35 e⁻/Å²/s dose rate for the dataset of HA trimer. 3861 micrographs were collected over 4.5 s with 11.31 e⁻/Å²/s dose rate for the dataset of HA trimer with MSBP. 1053 micrographs were collected over 4.5 s with 11.24 e⁻/Å²/s dose rate for the dataset of catalase. 1022 micrographs were collected over 4.5 s with 11.24 e⁻/Å²/s dose rate for the dataset of catalase with MSBP. 1317 micrographs were collected over 4.5 s with 11.24 e⁻/Å²/s dose rate for the dataset of β-galactosidase. 1278 micrographs were collected over 4.5 s with 11.25 e⁻/Å²/s dose rate for the dataset of β-galactosidase with MSBP.

All micrographs with 40 frames were collected with defocus values from −2.5 µm to −1.3 µm (for apoferritin and HA trimer) or −2.5 µm to −1.0 µm (for catalase and β-galactosidase).

Drift correction of the micrographs was performed using MotionCor2[34]. Motion-corrected sums without does-weighting were used for contrast transfer function (CTF) estimation with GCTF[35]. Motion-corrected sums with does-weighting were used for all other image processing. Particles picked by Gautomatch (https://www.mrclmb.cam.ac.uk/kzhang/Gautomatch/) were extracted by RELION[36] and then imported into CryoSPARC[37] for further processing. The initial model was generated using ab initio reconstruction. Well-sorted particles by iterative 2D classification and heterogeneous refinement were finally subjected to homogenous and non-uniform refinement to generate the final maps. CTF refinement was performed to further improve the final resolution. Detailed information of data processing of apoferritin can be found in Supplementary Fig. 1.

Similar approaches were used to process HA trimer (Supplementary Fig. 13a), catalase (Supplementary Fig. 2a, b), and β-galactosidase (Supplementary Fig. 2c, d) dataset. For HA trimer, 2D averages were used to evaluate the angular distribution of the particles. The final 2D classification results for HA trimer with or without MSBP can be found in Supplementary Fig. 13d, e. 2D templates of HA trimer with MSBP generated by a 3D density map were used to pick more particles from images for further processing to generate final map. Detailed information of data processing of HA trimer with MSBP can be found in Supplementary Fig. 13a. For the catalase subtraction datasets, the density was adjusted manually in the USCF-Chimera to generate the mask for monomer, dimer, and trimer. The particles were subtracted in cryoSPARC followed by local refinement. To increase the particle number for monomers, the symmetry expansion was applied with D2 symmetry followed by mask generation, subtraction, and local refinement.

For cryo-ET data of apoferritin, tilt series from -60° to 60° with 3° increments, were collected with a K3 Summit direct electron detector (Gatan) using Tomography (Thermo Fisher Scientific) in counting mode at a calibrated magnification of 42,000X (2.09 Å physical pixel size). A total dose of 2.16 e⁻/Å² (for apoferritin only and MSBP only), 2.92 e⁻/Å² (for apoferritin with MSBP) and 2.72 e⁻/Å² (for apoferritin with PEG and apoferritin with $Pd(NO_3)_2$) was used to expose every micrograph for around 1 s (4 frames per image) with a defocus near -4 µm.

For cryo-ET data of HA trimer, tilt series from -45° to 45° with 3° increments, were collected with a K3 Summit direct electron detector (Gatan) using Tomography (Thermo Fisher Scientific) in counting mode at a calibrated magnification of 53,000X (1.7 Å physical pixel size). A total dose of 4.19 e⁻/Å² (for HA trimer with and without MSBP) was used to expose every micrograph for around 1.8 s (6 frames per image) with a defocus near -5 µm. Similar approaches were applied for the cryo-ET of catalase, β-galactosidase, IMP1, IMP2, and CSW complex.

The micrographs were aligned by MotionCor2 without CTF correction. The tilt series were aligned based on gold fiducials by making seed model manually in Etomo[38]. After the generation of tomograms, the particles were picked by template matching in Dynamo[39].

**Reporting summary**. Further information on research design is available in the Nature Portfolio Reporting Summary linked to this article.

## Data availability
Cryo-EM density maps of tested proteins with MSBP applied have been deposited in the Electron Microscopy Data Bank under accession code EMD-36313 (Apoferritin), EMD-

36314 (Hemagglutinin), EMD-36315 (Catalase), and EMD-36316 (β-Galactosidase), respectively. The source data behind the graphs in Supplementary Fig. 17 can be found in Supplementary Data 1. All other data and materials are available from the corresponding authors upon reasonable request. Source data are provided in this paper.

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

## Acknowledgements
We thank Bik Tye and Yifan Cheng for their critical reading of the manuscript, Masa Kikkawa at the University of Tokyo for generously sharing mouse heavy chain apo-ferritin plasmid, Shiqian Qi at Sichuan University for sharing CSW complex protein with us. We also thank Chunlin He, Zhongguang Yang, and Fei Sun for their helpful discussion in the early stage of this project. All cryo-EM data were collected at the Biological Cryo-EM Center at HKUST, generously supported by a donation from the Lo Kwee Seong Foundation. This project is supported by grants of S.D. from Hong Kong Research Grants Council (ECS26101919, GRF16103321, GRF16102822, C7009-20GF, C6001-21EF), Southern Marine Science and Engineering Guangdong Laboratory (Guangzhou) (SMSEGL20SC01-L), Guangdong Basic and Applied Basic Research Foundation (2021A1515012460), Shenzhen Special Fund for Local Science and Technology Development Guided by Central Government (2021Szvup140), HKUST VPRDO 30 for 30 Research Initiative Scheme, and HKUST start-up and initiation grants. Y.W. acknowledges support from the General Research Fund (17308518) and Collaborative Research Fund (C7075-21G) from the Hong Kong Research Grants Council.

## Author contributions
Y.X., Y.Q., and S.D. designed and performed single-particle cryo-EM experiments. Y.X., Y.Q., and Y.Z. performed cryo-ET experiments. Y.X. Y.Q. and L.W. prepared specimens using MSBP. S.D. and Y.W. supervised experiments and data analysis. All authors contributed to manuscript preparations.

## Competing interests
The authors declare the following competing interests: the method applying metallo-supramolecular branched polymer (MSBP) in sample preparation of single particle cryo-electron microscopy (cryo-EM) is described in a provisional patent application.
