## [Peer Review File · Communications Biology]

Reviewers' comments:

Reviewer #1 (Remarks to the Author):

In this manuscript, the authors described a method for cryo-EM sample preparation on regular holey carbon grids. The protein sample is mixed with PEGs and metal salts, which form metallo-supramolecular branched polymer (MSBP). The MSBP particles are attracted to occupy the air-water-interface (AWI) and loosely bonded to the proteins due to electrostatic interaction. The polymer apparently changes the viscosity of the protein solution. The synergetic effects lead to the proteins being kept away from the AWI.

Overall I consider this method useful and am convinced from the data that proteins are indeed kept in the center of vitreous ice. However I suggest the authors adopt a softer tone regarding whether this method can improve the resolution. Denaturation of protein and preferred orientation are common sources for deteriorated resolution but not the full picture. Mitigating proteins from the AWI may change orientation distribution, but not necessarily lead to improved resolution. In my opinion, this method, just like all others mentioned in the manuscript, such as adding a support layer to the grids and pre-mixing with surfactants, are all viable routes if researchers suspect their resolution is limited by the AWI issue. It will be too strong a claim that a single approach can resolve the orientation bias, let alone the resolution enhancement.

I share with both reviewers the concern regarding polymer particles at the AWI. Although the authors showed the noises did not affect the processing and I do believe this issue can be resolved, there are questions un-answered. Unlike the M12L24 taken as a control in the response letter, the MSBP used in this work is not well characterized. What are the structure/molecular formula of the polymer particles? Are they self-assembled into a 2-D network or are they isolated particles? The authors ought to take into consideration that the MSBP used here is not molecular surfactants, properties of which are available to researchers. Most of them will be unlikely to choose MSBP, since it will introduce a new uncontrollable factor unless, in a desperate scenario.

I will recommend publication if the issues above are addressed.

Reviewer #2 (Remarks to the Author):

The authors present the use of metallo-supramolecular branched polymers (MSBP) to alleviate the effects of air-water interface in cryo-EM. Air-water interface effects is a broad term that refers to a situation hypothesised in cryo-EM where proteins attach to the surface of the liquid on the TEM grid and are denatured or assume fixed orientations on that surface. The results include TEM images where all or most of the particles are damaged or the lack of certain orientations of the molecule means that a good 3D structure is difficult or impossible to achieve. In my opinion this might be one of the biggest problems, if not the biggest problem, facing the more wide applicability of the single-particle cryo-EM technique. Therefore a widely applicable solution to this issue is of great import to cryo-EM and structural biology as a whole.

Broadly speaking, the authors present four different sets of results. The first is cryo-EM structures of apo-ferritin, a popular cryo-EM standard, to demonstrate that high-resolution is still possible with MSBP added to the solution - in fact a higher resolution is achieved with MSBP. The second is catalase particles, which are much smaller than apo-ferritin which demonstrate the applicability of the technique to smaller proteins. Often an issue with cryo-EM methods papers is that only apo-ferritin

data is presented and the ease with which this protein is solved with modern cryo-EM gives a misleading impression as to the broader utility of the advance for other types of sample. The third set of experimental results contains tomography reconstructions which show that the additive improves the depth distribution of particles within the ice - with MSBP added particles are more likely to be found in the bulk of the ice layers rather than sticking to the surface - a situation which correlates with air-water interface effects. The fourth set of results demonstrates improvement for a particle, hemagglutinin, that is known to demonstrate air-water interface problems. On the whole these datasets make a compelling case that MSBP does not have a large deleterious effect on resolution for large and small particles, improves the distribution of particles within the ice and effectively addresses preferred orientation. I think the paper can be published with some minor revisions which I list below.

1. The authors hypothesise that the MSBP surrounds particles and "traps" them away from the air-water interface - I suspect that a more likely explanation is that the MSBP forms a "coverslip" on the air-water interface that limits or blocks the protein's interaction with the interface. Similar behaviours with apo-ferritin proteins and detergents are hypothesised by Rob Glaeser in:

Glaeser, Robert M. "Proteins, interfaces, and cryo-EM grids." *Current opinion in colloid & interface science* 34 (2018): 1-8.

This suspicion is informed by the visibility of MSBP particles on the AWI in the tomography results - if the authors can provide multiple examples in the tomograms where the proteins are attached to MSBPs they will convince me of their own hypothesis

2. The solution of the catalase monomer is important to the author's case that MSBP does not affect cryo-EM of small particles. To increase the number of catalase monomers in processing the authors pursue a particle subtraction and symmetry expansion process that I find unconvincing. The result in Fig. 2(a) where the monomer is solved to ~ 3.5 Å is sufficiently convincing and the subtraction and symmetry expansion should be removed.

3. In previous review there was some controversy about the extra noise that would be introduced into the micrographs with the added MSBP. I am less concerned about this than the previous set of reviewers, other approaches to tackle AWI effects such as graphene oxide and detergents can introduce worse artefacts but like to see magnified versions of supplementary Fig. 15, which shows raw micrographs for each single particle dataset. The MSBP particles are clearly visible in sub-figure (a). In sub-figures (b) and (c) it appears there are less intact particles in the MSBP case. I would also like to see a histogram of the number of picked particles that make it to the final map per micrograph for each dataset to test my theory as to whether there are fewer actual particles or if this is just my impression based off of 1 micrograph.

4. Supplementary figure 4 is too low resolution to see the necessary detail. It would also help with interpretation if the z positions of each slice were indicated on a z-x cross section of the tomogram.

5. There is some discussion about the fact that local CTF refinement is especially beneficial for the apo-ferritin result in supplementary figure 3 - this is clearly due to the more uniform dispersion of particles within the ice as seen in the tomography results and the authors can shorten the discussion of this fact. They should annotate the improvements due to local and global CTF refinement on the relevant figure.

6. Line 48 "Cryo-EM have powered a "resolution revolution" in structural biology by circumventing the major challenges faced when using traditional X-ray crystallography method" needs to be split into two sentences since it implies that resolution of cryo-EM has surpassed crystallography, whereas in reality

the revolution has made cryo-EM a viable alternative to crystallography.

7. Line 54 " vitrification of specimen is a delicate process" I think the authors are referring to the whole sample prep process - vitrification just refers to the rapid freezing of grids to achieve vitreous or amorphous ice.

The ultimate impact of the paper will depend on its applicability to many different protein samples. Other approaches to solving preferred orientation have proved incredibly useful for some proteins but not for others. Previous reviewers have criticized the current manuscript for not having a wider range of protein samples but I regard this suggestion as being beyond the scope of the paper - it is for the wider cryo-EM field to try the author's approach to their own samples and see if it helps to deal with challenging proteins.

Response to Reviewers' comments

Reviewer #1 (Remarks to the Author):

In this manuscript, the authors described a method for cryo-EM sample preparation on regular holey carbon grids. The protein sample is mixed with PEGs and metal salts, which form metallo-supramolecular branched polymer (MSBP). The MSBP particles are attracted to occupy the air-water-interface (AWI) and loosely bonded to the proteins due to electrostatic interaction. The polymer apparently changes the viscosity of the protein solution. The synergetic effects lead to the proteins being kept away from the AWI.

Overall I consider this method useful and am convinced from the data that proteins are indeed kept in the center of vitreous ice. However I suggest the authors adopt a softer tone regarding whether this method can improve the resolution. Denaturation of protein and preferred orientation are common sources for deteriorated resolution but not the full picture. Mitigating proteins from the AWI may change orientation distribution, but not necessarily lead to improved resolution. In my opinion, this method, just like all others mentioned in the manuscript, such as adding a support layer to the grids and pre-mixing with surfactants, are all viable routes if researchers suspect their resolution is limited by the AWI issue. It will be too strong a claim that a single approach can resolve the orientation bias, let alone the resolution enhancement.

We thank the reviewer for providing positive feedback and valuable constructive suggestions to improve our manuscript. We acknowledge the reviewer's suggestion regarding the need for caution and modesty when discussing the potential resolution improvement through the application of MSBP. In response, we have made the necessary revisions in the manuscript to adopt a more tempered tone in the description of this aspect (Lines 151-160, 393).

I share with both reviewers the concern regarding polymer particles at the AWI. Although the authors showed the noises did not affect the processing and I do believe this issue can be resolved, there are questions un-answered. Unlike the M12L24 taken as a control in the response letter, the MSBP used in this work is not well characterized. What are the structure/molecular formula of the polymer particles? Are they self-assembled into a 2-D network or are they isolated particles? The authors ought to take into consideration that the MSBP used here is not molecular surfactants, properties of which are available to researchers. Most of them will be unlikely to choose MSBP, since it will introduce a new uncontrollable factor unless, in a desperate scenario.

I will recommend publication if the issues above are addressed.

We thank the reviewer for pointing this out. The MSBPs are isolated local clusters consisting of palladium and PEG polymers through coordination bonds. While the MSBPs themselves have not been directly characterized, their structure can be inferred by a few experiments. Firstly, when the same polymer is present at a higher concentration, it forms a 3D network, i.e., a gel¹. In the case of MSBPs, they are formed at a significantly lower concentration, below the threshold required for gel formation. Secondly, upon the formation of MSBPs, the viscosity of the solution increases, as indicated in Supplementary Video 1. This viscosity increase indicates

the formation of higher molecular species—in this case the local polymer clusters. Thirdly, based on the control experiment of M12L24, which are well-defined polymer particles, we can deduce that MSBPs share a similar nature, although their size may not be uniform. In fact, thermal annealing is needed to form M12L24 structure. According to a previous simulation study, local cluster of different sizes are formed prior to annealing². Since MSBPs can already work for our purpose, there is no need for thermal annealing, which would complicate the process. This implies that by simply mixing the polymer and palladium, the MSBPs are readily available for use.

We understand that there may be hesitation in using the MSBP. However, it is important to note that both PEG and palladium have been verified as biologically friendly reagents in previous studies (Lines 270-276). As mentioned earlier, the MSBP is user-friendly and can be conveniently employed, for instance, by creating it as a kit that only requires simple mixing. Furthermore, we have demonstrated that proteins treated with MSBP can achieve near-atomic resolution in multiple distinct structures. Consequently, the MSBP presents an additional alternative method (and an alternative mechanism) for cryo-EM sample preparation.

Reviewer #2 (Remarks to the Author):

The authors present the use of metallo-supramolecular branched polymers (MSBP) to alleviate the effects of air-water interface in cryo-EM. Air-water interface effects is a broad term that refers to a situation hypothesised in cryo-EM where proteins attach to the surface of the liquid on the TEM grid and are denatured or assume fixed orientations on that surface. The results include TEM images where all or most of the particles are damaged or the lack of certain orientations of the molecule means that a good 3D structure is difficult or impossible to achieve. In my opinion this might be one of the biggest problems, if not the biggest problem, facing the more wide applicability of the single-particle cryo-EM technique. Therefore a widely applicable solution to this issue is of great import to cryo-EM and structural biology as a whole.

Broadly speaking, the authors present four different sets of results. The first is cryo-EM structures of apo-ferritin, a popular cryo-EM standard, to demonstrate that high-resolution is still possible with MSBP added to the solution - in fact a higher resolution is achieved with MSBP. The second is catalase particles, which are much smaller than apo-ferritin which demonstrate the applicability of the technique to smaller proteins. Often an issue with cryo-EM methods papers is that only apo-ferritin data is presented and the ease with which this protein is solved with modern cryo-EM gives a mis-leading impression as to the broader utility of the advance for other types of sample. The third set of experimental results contains tomography reconstructions which show that the additive improves the depth distribution of particles within the ice - with MSBP added particles are more likely to be found in the bulk of the ice layers rather than sticking to the surface - a situation which correlates with air-water interface effects. The fourth set of results demonstrates improvement for a particle, hemagglutinin, that is known to demonstrate air-water interface problems. On the whole these datasets make a compelling case that MSBP does not have a large deleterious effect on resolution for large and small particles, improves the distribution of particles within the ice and effectively addresses preferred orientation. I think the paper can be published with some minor revisions which I list below.

Thank the reviewer for the positive comments. The guidance provided will significantly contribute to improving the quality of our manuscript.

1. The authors hypothesise that the MSBP surrounds particles and "traps" them away from the air-water interface - I suspect that that a more likely explanation is that the MSBP forms a "coverslip" on the air-water interface that limits or blocks the protein's interaction with the interface. Similar behaviours with apo-ferritin proteins and detergents are hypothesised by Rob Glaesar in:

Glaeser, Robert M. "Proteins, interfaces, and cryo-EM grids." *Current opinion in colloid & interface science* 34 (2018): 1-8.

This suspicion is informed by the visibility of MSBP particles on the AWI in the tomography results - if the authors can provide multiple examples in the tomograms where the proteins are attached to MSBPs they will convince me of their own hypothesis.

Thank the reviewer for pointing this out. We agree with the reviewer's assessment that the term "trap" is not suitable for describing the situation in question. Therefore, we have replaced the word "trap" with "shield" or "protect" (Lines 87, 261 and 384) and appropriately cited the relevant paper. In the discussion section, we propose the first possible mechanism that "MSBP may occupy the AWI, thereby shielding protein particles away from the AWI." (Lines 295-305). The role of MSBP in shielding particles is akin to that of a coverslip.

2. The solution of the catalase monomer is important to the author's case that MSBP does not affect cryo-EM of small particles. To increase the number of catalase monomers in processing the authors pursue a particle subtraction and symmetry expansion process that I find unconvincing. The result in Fig. 2(a) where the monomer is solved to ~3.5 Å is sufficiently convincing and the subtraction and symmetry expansion should be removed.

Thanks for the thoughtful concern. The catalase monomer, without symmetry expansion, has achieved a satisfactory resolution, which could be further improved by utilizing a larger dataset. As a result, we have removed the Fig.2(b) from the main figure to the supplementary figure 6.

3. In previous review there was some controversy about the extra noise that would be introduced into the micrographs with the added MSBP. I am less concerned about this than the previous set of reviewers, other approaches to tackle AWI effects such as graphene oxide and detergents can introduce worse artefacts but like to see magnified versions of supplementary Fig. 15, which shows raw micrographs for each single particle dataset. The MSBP particles are clearly visible in sub-figure (a). In sub-figures (b) and (c) it appears there are less intact particles in the MSBP case. I would also like to see a histogram of the number of picked particles that make it to the final map per micrograph for each dataset to test my theory as to whether there are fewer actual particles or if this is just my impression based off of 1 micrograph.

We thank the reviewer for the positive feedback and valuable suggestions. We acknowledge that the background noise introduced by MSBP is within acceptable limits, and its impact can be further mitigated by collecting a larger dataset. We agree

with the reviewer's assessment that a smaller number of particles are observed in the micrograph when MSBP is applied, partially due to the background noise. On the other hand, we used the same protein concentration for cryo-sample preparation, regardless of the presence or absence of MSBP. In the case of cryo-sample with MSBP, there is a greater spatial distribution available for protein molecules, as MSBP pushes particles towards the central layer of the vitreous ice. Consequently, fewer particles are observed in the raw micrograph of the MSBP case compared to the micrograph of the protein alone. To verify this hypothesis, we have calculated the average number of particles per micrograph and included the results in Supplementary Figure 17, as suggested by the reviewer. In most cases, the protein with MSBP exhibited a slightly lower particle count compared to the protein alone, both for the G-automatch picked particles and the particles in the final reconstructed map. Additionally, we have included magnified views of each micrograph to clearly display the particles.

Supplementary Fig. 17 Statistics of particle numbers per micrograph with and without MSBP. The distribution of particle numbers per micrograph is shown, with MSBP represented in orange and without MSBP represented in blue. The analysis includes apoferritin, HA-trimer, catalase, and β -Galactosidase. For individual protein samples, the left side shows the G-automatch picked particles, while the right side displays the particles used for the final reconstruction.

4. Supplementary figures 4 is too low resolution to see the necessary detail. It would also help with interpretation if the z positions of each slice were indicated on a z-x cross section of the tomogram.

Thanks for pointing this out. The low resolution of the image resulted from the conversion of the .doc file to a .pdf file. We have addressed this issue and have provided a high-resolution image that includes the necessary level of detail. Additionally, we have indicated the z position of each slice at the top of the figure to enhance clarity (Supplementary Figure 4).

5. There is some discussion about the fact that local CTF refinement is especially beneficial for the apo-ferritin result in supplementary figure 3 - this is clearly due to the more uniform dispersion of particles within the ice as seen in the tomography results

and the authors can shorten the discussion of this fact. They should annotate the improvements due to local and global CTF refinement on the relevant figure.

Thanks for the reviewer's suggestion. We have revised and condensed the discussion regarding the CTF refinement (Lines 151-161). The enhancements resulting from both local and global CTF refinement have been clarified and illustrated in Supplementary Figure 3b.

6. Line 48 "Cryo-EM have powered a "resolution revolution" in structural biology by circumventing the major challenges faced when using traditional X-ray crystallography method" needs to be split into two sentences since it implies that resolution of cryo-EM has surpassed crystallography, whereas in reality the revolution has made cryo-EM a viable alternative to crystallography.

We thank the reviewer for raising this thoughtful concern. We have made the following modifications to the expressions in order to avoid misinterpretation: "Recent technical breakthroughs in single-particle cryo-EM have powered a "resolution revolution" in structural biology. Cryo-EM circumvents the major challenges faced when using the traditional X-ray crystallography method. " (Lines 47-50).

7. Line 54 " vitrification of specimen is a delicate process" I think the authors are referring to the whole sample prep process - vitrification just refers to the rapid freezing of grids to achieve vitreous or amorphous ice.

We apologize for the misinterpretation. We have removed the statement "because vitrification of specimen is a delicate process". The revised expression now refers to the entire process of cryo-specimen preparation (Lines 51-54).

The ultimate impact of the paper will depend on its applicability to many different protein samples. Other approaches to solving preferred orientation have proved incredibly useful for some proteins but not for others. Previous reviewers have criticized the current manuscript for not having a wider range of protein samples but I regard this suggestion as being beyond the scope of the paper - it is for the wider cryo-EM field to try the author's approach to their own samples and see if it helps to deal with challenging proteins.

We sincerely thank the reviewer for the positive feedback and for recognizing the significance of our study.

Reference:

1. Zhukhovitskiy, A. V. *et al.* Highly branched and loop-rich gels via formation of metal-organic cages linked by polymers. *Nat. Chem.* **8**, 33–41 (2016).
2. Yoneya, M., Tsuzuki, S., Yamaguchi, T., Sato, S. & Fujita, M. Coordination-directed self-assembly of M12L24 nanocage: Effects of kinetic trapping on the assembly process. *ACS Nano* **8**, 1290–1296 (2014).

REVIEWERS' COMMENTS:

Reviewer #1 (Remarks to the Author):

I am happy with the revised version of the manuscript and recommend publication.

Reviewer #2 (Remarks to the Author):

I am satisfied that the authors have addressed the minor points I raised in my earlier review and can now recommend this paper for publication.